# Neuroprotective Effects of Activated Protein C Involve the PARP/AIF Pathway against Oxygen-Glucose Deprivation in SH-SY5Y Cells

**DOI:** 10.3390/brainsci10120959

**Published:** 2020-12-10

**Authors:** Mukesh Kumar Sriwastva, Remesh Kunjunni, Mutahar Andrabi, Kameshwar Prasad, Renu Saxena, Vivekanandhan Subbiah

**Affiliations:** 1Department of Neurobiochemistry, All India Institute of Medical Sciences, New Delhi 110029, India; remeshmarangad@gmail.com (R.K.); mutahar.andrabi@gmail.com (M.A.); vivek.neurobiochemistry@gmail.com (V.S.); 2Department of Neurology, All India Institute of Medical Sciences, New Delhi 110029, India; kp0704@gmail.com; 3Department of Hematology, All India Institute of Medical Sciences, New Delhi 110029, India; renusaxena@outlook.com

**Keywords:** neuroprotection, activated protein C, cerebral ischemia, oxygen-glucose deprivation

## Abstract

Protein C, a member of the zymogen family of serine proteases in plasma, is one of the several vitamin K dependent glycoproteins known to induce anti-apoptotic activity. However, the target molecule involved in the mechanism needs to be investigated. We sought to investigate the pathways involved in the anti-apoptotic role of activated protein C (APC) on oxygen-glucose deprivation (OGD) induced ischemic conditions in in-vitro SH-SY5Y cells. SH-SY5Y cells were exposed to OGD in an airtight chamber containing 95% N_2_ and 5% CO_2_ and media deprived of glucose for 4 h following 24 h of reoxygenation. The cell toxicity, viability, expression of receptors such as endothelial cell protein C receptor (EPCR), protease-activated receptor (PAR)1, PAR3, and apoptosis-related proteins B-cell lymphoma 2 (BCL-2), BCL-2-like protein 4 (Bax), Poly [ADP-ribose] polymerase-1 (PARP-1) were assessed. Administration of APC decreased the cellular injury when compared to the OGD exposed group in a dose-dependent manner and displayed increased expression of PAR-1, PAR-3, and EPCR. The APC treatment leads to a reduction in PARP-1 expression and cleavage and apoptosis-inducing factor (AIF) expression. The reduction of caspase-3 activity and PARP-1 and AIF expression following APC administration results in restoring mitochondrial function with decreased cellular injury and apoptosis. Our results suggested that APC has potent protective effects against in-vitro ischemia in SH-SY5Y cells by modulating mitochondrial function.

## 1. Introduction

Ischemic stroke is the second leading cause of death and the leading cause of long-term adult disability worldwide [1]. A number of clinical trials have been carried out to investigate the potential neuroprotective compounds of ischemic strokes [2]. The researchers have targeted the development of multi-target strategies to protect neuronal cells following ischemic injury. Protein C, a member of the zymogen family of serine proteases in plasma, is one of several vitamin K dependent glycoproteins. Cleavage of protein C at Arg169 by thrombin removes the activation peptide and generates activated protein C (APC) [3]. The anticoagulant characteristics of APC have been well-studied [4,5]. The Protein C Evaluation in Severe Sepsis (PROWESS) study showed that recombinant human activated protein C (rhAPC) decreased mortality in severe sepsis patients by 19.4% [6]. Low levels of plasma protein C are a risk factor for ischemic stroke incidents [7], and rhAPC modulates several genes in the endothelial apoptosis pathway, including the B-cell lymphoma 2 (BCL-2) homologous protein and inhibitor of apoptosis protein. These pathway changes result in the ability of rhAPC to inhibit the induction of apoptosis [8]. APC has shown anti-apoptotic activity in human endothelial cell lines and transformed kidney cells [9]. It was shown that APC downregulates the expression of nuclear factor-κB (NF-κB)-dependent pro-inflammatory genes, directly provides anti-apoptotic activity in a dose-dependent manner by inhibiting the transcription of tumor suppressor protein p53 and pro-apoptotic BCL-2-like protein 4 (Bax)/BCL-2 ratio, and leads to the reduction of caspase-3 signaling in endothelial cells [10]. It exerts its anti-apoptotic activity by binding to the endothelial protein C receptor (EPCR) and signaling via a protease-activated receptor (PAR)1 and PAR3 on endothelial cells [11,12,13,14,15,16,17].

Poly(ADP-ribose) polymerase 1 (PARP-1) is a family of nuclear enzymes with diverse functions in chromatin structure, transcription, and genomic integrity, and it plays a role in the inflammatory pathogenesis of many central nervous system disorders including stroke [18,19]. PARP-1 involves translocation of apoptosis-inducing factor (AIF) from mitochondria to the nucleus in response to DNA damage [20]. PARP-1 mediated cell death is another important form of programmed necrosis [21]. In pathophysiological conditions such as ischemic stroke, the over-activation of PARP-1 exerts deleterious effects, as demonstrated in several experimental models of cerebral ischemia [22,23]. In rodent models of cerebral ischemia, studies have shown that PARP inhibitors reduce infarct volume, blood-brain barrier permeability, brain edema, spontaneous and recombinant tissue plasminogen activator (rtPA) induced hemorrhagic transformations, inflammatory response, and motor deficit and enhances long-term neuronal survival and neurogenesis [24,25,26].

In this study, we applied an in-vitro oxygen-glucose deprivation (OGD) model of SH-SY5Y cell cultures to investigate APC neuroprotective activity against OGD (ischemic insult) and its effect on the molecular pathways. The influence of APC on the expression of PARP-1 expression and mitochondrial membrane potential were investigated in in-vitro SH-SY5Y cells exposed to OGD.

## 2. Materials and Methods

### 2.1. Cell Culture and Treatment

The SH-SY5Y neuroblastoma cell line was procured from the National Centre for Cell Sciences, Pune, India, and maintained in Dulbecco’s Modified Eagle’s medium (DMEM) (Sigma–Aldrich, St. Louis, MO, USA) and Ham’s F12 nutrient mixture (F12) (Sigma–Aldrich, USA) in 1:1 ratio. The media were supplemented with 2 mmol L-glutamine, 10% heat-inactivated fetal bovine serum (FBS), and 1% antibiotics (penicillin/streptomycin) at 37 °C in a humidified CO_2_-incubator containing 5% CO_2_, and the medium was changed at 2–3 days intervals. Differentiation of SH-SY5Y cells into mature neuron-like phenotype was carried out by supplementing 10 μM retinoic acid for 5 days (Sigma–Aldrich, USA) and confirmed by the expression of neuron-specific microtubule-associated protein 2 (MAP2) and tyrosine hydroxylase markers by fluorescent microscopy.

To perform oxygen-glucose deprivation (OGD), SH-SY5Y cells underwent serum starvation for 4 h before OGD exposure, which helps in maintaining low oxygen concentration during OGD treatment [27,28]. Cells were then exposed to combined OGD in an airtight humidified hypoxic chamber (Stemcell Technologies, Vancouver, BC, Canada) pre-gassed with an anoxic gas mixture of 5% CO_2_ and 95% N_2_ for 5 min (min) [29]. The OGD was carried out in Dulbecco′s modified—eagle′s medium (DMEM) media deprived of glucose and FBS and bubbled with 5% CO_2_/95% N_2_ gas for 30 min. Cells were exposed to OGD for different time interval (1, 2, 3, 4, 5, and 6 h) and were placed back in the normoxic condition with a normal culture medium. For concurrent treatment, human APC (Haematologic Technologies Inc., Essex, VT, USA) was present in the culture medium during OGD and reoxygenation; for the control group, only the vehicle (50% Glycerol/H_2_O) was added to the culture medium.

### 2.2. Cell Viability Assessment

Following 24 h reoxygenation, post OGD cells were used for viability assay using Cell Counting Kit-8 (CCK-8, Sigma–Aldrich, USA) and lactate dehydrogenase (LDH) assay (BioVision, Inc., Milpitas, CA, USA) according to the manufacturer’s instructions. The results are expressed as the percentage of 3-(4,5-dimethylthiazol-2-yl)-2,5-diphenyl tetrazolium bromide (MTT) reduction and a percentage of LDH release into the culture medium relative to the control cells for CCK-8 and LDH assay, respectively. All assays were performed in triplicate.

Acridine orange (AO)/ethidium bromide (EB) staining was used to assess morphology and to measure apoptotic and necrotic cells [30]. The cells were harvested and re-suspended in phosphate-buffered saline (PBS). Aliquots of cell suspension (20 µL) were stained with the dye mixture (4 µL; acridine orange (100 mg/mL) and ethidium bromide (100 mg/mL)). The examination was done using a Nikon inverted fluorescence microscope (Nikon Eclipse Ti-S) at 40× magnification. The number of cells in each group was expressed as a percentage of the total cell number, and a minimum of 300 cells was counted per treatment in duplicate.

### 2.3. Electron Microscope Assay

Cells were fixed for 4 h in Karnovskys fixative (4% paraformaldehyde and 2% glutaraldehyde solution) and transferred to a phosphate buffer and subsequently processed for transmission electron microscopy (TEM). The cells were postfixed in 1% osmium tetroxide (Sigma–Aldrich, USA) for 2 h at 4 °C. The cells were then dehydrated in grades of acetone cleared in toluene and infiltrated with Araldite resin, and blocks were prepared by embedding in Araldite. After polymerization of the embedding medium, ultra-thin sections were collected on metal grids, which were stained using uranyl acetate and lead citrate (Sigma–Aldrich, USA). After washing in distilled water, the grids were stored in airtight Petri plates for viewing under the TEM.

### 2.4. PAR1, PAR3, and EPCR Expression by Flow Cytometry

The cells were harvested 24 h post OGD exposure, washed, and re-suspended in staining buffer (phosphate-buffered saline, 0.2% bovine serum albumin, pH-7.2). The 50 μL (2 × 10^6^ cells/mL) aliquot of each cell suspension was stained with 50 μL of each fluorochrome-conjugated antibody: PAR1—Phycoerythrin (PE) (Bioss Antibodies, Woburn, MA, USA), PAR3-fluorescein isothiocyanate (FITC) (Bioss Antibodies, USA), and EPCR-Allophycocyanin (BioLegend, San Diego, CA, USA) for 30 min in the dark at 4 °C. One milliliter of PBS buffer was added to each tube and centrifuged at 500× *g* for 5 min. Cells were analyzed by flow cytometry on a BD FACSCantoTM flow analyzer (BD Biosciences, San Jose, CA, USA) equipped with analysis software.

### 2.5. Annexin V/Propidium Iodide (PI) Double-Staining Assay

The SH-SY5Y cells were harvested and collected by centrifugation at 500× *g* for 10 min at 4 °C. The cells were washed twice with cold PBS and then re-suspended in 1X binding buffer (0.01 M HEPES, pH 7.4; 0.14 M NaCl; 0.25M CaCl_2_) at a concentration of ~1 × 10^6^ cells/mL. A 100 µL sample of the cell suspension (~1 × 10^5^ cells) was transferred to a 15 mL conical tube. Annexin V—Allophycocyanin conjugated (5 µL) (Thermo Fisher Scientific, Waltham, MA, USA) and PI (2 µL of 1 mg/mL) (Himedia, Bengaluru, India) were added to the cell suspension, mixed, and incubated for 15 min at room temperature (RT) in the dark. A sample of 400 µL of 1X binding buffer was added to each tube and analyzed by flow cytometry within 1 h. Flow cytometric analysis of cell suspensions was carried out by using BD FACSCantoTM (excitation 520 nm and emission 620 nm). Data for 20,000 events were analyzed for each experiment using BD FACS Diva software (version 6.1.3).

### 2.6. Measurement of Mitochondrial Membrane Potential (MMP)

The dynamics of the loss of MMP were examined by cytofluorimetric, lipophilic cationic dye, 5,5′,6,6′-tetrachloro-1,1′,3,3′-tetraethylbenzimi-dazolylcarbocyanine iodide (JC-1) using flow cytometry and fluorescent microscope. Following the respective treatments, cells were harvested and 100 μL of the JC-1 Staining Solution (Cayman Chemical Company, Ann Arbor, MI, USA) were added per ml of cell suspension (8 × 10^4^ cells/mL) and incubated for 30 min. The samples were directly analyzed by flow cytometry and fluorescent microscopy. The healthy cells with functional mitochondria contain red JC-1 J-aggregates. Apoptotic or unhealthy cells with collapsed mitochondria contain mainly green JC-1 monomers.

### 2.7. Caspase-3 Activity Assay

The caspase-3 activity was determined using a caspase-3 kit (Abcam Biotech Company, Cambridge, UK) as per the manufacturer’s instruction. Briefly, cells were lysed after their respective treatments using the lysis buffer provided by the kit. The cells were re-suspended in 50 µL of the cell lysis buffer. Assays were performed in 96-well Microtiter plates by adding 50 µL of the reaction buffer to the 50 µL cellular extracts and 5 µL of the 4 mM DEVD-p-NA substrate. The reaction mixture was incubated for 1 h at 37 °C, and the caspase-3 activity was determined at an absorbance of 405 nm with a microplate reader (Bio-Rad, Hercules, CA, USA). Results were expressed as fold changes compared with the control and OGD group.

### 2.8. Analysis of BCL-2, Bax, PARP-1 and AIF mRNA Expression

Total RNA was extracted with a PureLink^®^ RNA Mini Kit (Ambion by Life Technologies, Carlsbad, CA, USA) according to the manufacturer’s instructions, and RNA purity and integrity were tested by an optical density (OD) ratio between 1.8 and 2.0 and by gel electrophoresis, respectively. For each group, cDNA was synthesized from 500 ng of RNA using a Maxima Universal First Strand cDNA Synthesis kit (Thermo Scientific, Waltham, MA, USA, catalog no-K1661), and consecutively 50 ng of cDNA was used for quantitative real-time PCR (qRT-PCR) assay. cDNA was amplified in a thermal cycle (CFX96 PCR system, Bio-Rad Laboratories, Hercules, CA, USA) in 20 µL reaction mixture containing 0.5 µL cDNA template (50 ng), 12.5 µL Maxima SYBR green/fluorescent qRT-PCR master mix (2X, Thermo Scientific), 0.3 µL forward and reverse primer (100 pmol), and 6.4 µL nuclease-free water. The details of the primer sequences are listed in Table 1. The data were normalized with housekeeping gene glyceraldehyde 3-phosphate dehydrogenase (GAPDH) (internal control), and expression was analyzed in terms of 2-∆Ct for control and treatment groups [31].

### 2.9. Western Blotting

The SH-SY5Y cells were seeded onto 6-well plates at a density of 2 × 10^5^ cells/well and exposed to OGD in the presence or absence of APC. Following 24 h of OGD, cells were harvested and lysed for 30 min at 4 °C with a cell lysis buffer containing a protease inhibitor cocktail (Thermo Scientific, USA). Protein concentrations were determined in each sample by Bradford assay using a microplate reader (iMark™ Microplate Absorbance Reader Bio-Rad, USA). Proteins were separated by 12% SDS polyacrylamide gel electrophoresis, and proteins were transferred to a nitrocellulose membrane (Bio-Rad, USA). The following primary antibodies were used: BCL-2 1:1000 (BioLegend, USA), Bax 1:1000 (BioLegend, USA), PARP-1 1:1000 (Santa Cruz Biotechnology, Inc., Dallas, TX, USA), and β-actin (Bioss Antibodies, USA). Proteins were detected by enhanced chemiluminescence reagents (Thermo Scientific, USA).

### 2.10. Statistical Analysis

All experimental data were expressed as the mean ± SD of three to four independent experiments and compared by one-way analysis of variance (ANOVA). The difference between the two treatments was considered significant at *p* ≤ 0.05. The statistical software SPSS 17.5 (SPSS, Chicago, IL, USA) was used.

## 3. Results

### 3.1. Effect of OGD Exposure in Differentiated SH-SY5Y Cells

The cell viability and the increase in LDH release were measured after OGD exposure for different time intervals (1, 2, 3, 4, 5, and 6 h) and 24 h of reoxygenation using CCK-8 assay and LDH assay, respectively. Significant reductions in cell viability (68.17% ± 2.4) and an increase in cell death was observed at 2 h OGD exposure compared to the control group (no OGD exposure), as depicted in Figure 1. A consecutive gradual decrease in cell viability percentage was continued in all further OGD periods (at 3, 4, 5, and 6 h); i.e., 55.42 ± 3.8%, 45.58 ± 1.7%, 38.34 ± 1.3%, and 33.50 ± 1.6%, respectively. The decrease in cell viability and increased cytotoxicity was significant with 4 h OGD compared with the control. As per the recommendations of international regulatory agencies, loss of cell viability of more than 50% is considered a cytotoxic response; thus, the OGD period of 4 h falls under this category (Figure 1A,B). Therefore, an OGD period of 4 h was selected for further experimentation to find out the neuroprotection effect of APC.

Examination of acridine orange/ethidium bromide staining revealed morphological changes in chromatin staining that distinguished apoptotic cells from normal healthy cells. The acridine orange/ethidium bromide staining of SH-SY5Y cells revealed that exposure to OGD for 4 h caused 43.19 + 8.9% of apoptosis compared to the control group, which was 5.28 + 4.05% (Figure 1C,D).

### 3.2. Effect of APC Treatment on OGD Induced Cell Death

First of all, the cytotoxicity effects of APC on differentiated SH-SY5Y cells were examined. No significant changes in cell survival, toxicity, and morphological changes were observed in APC treated SH-SY5Y cells. The effect of APC on OGD-induced hypoxic injury was determined. The SH-SY5Y cells were treated with APC (10–100 nM) or the vehicle during OGD and 24 h following OGD. As shown in Figure 2A, LDH release assay elucidated the protective role of APC; APC treatment decreases the LDH release in a dose-dependent manner and decreased up to 12.89 ± 1.50% (*p* ≤ 0.05, compare to OGD), which shows that APC significantly attenuates cell death against OGD induced neurotoxicity.

Transmission electron microscopy (TEM) reveals that OGD exposed cells cytoplasm contained many vesicles (red arrows) with the typical morphological features of autophagosomes and several isolated double or multi-membrane structures that engulfed cytoplasmic fractions and organelles. OGD groups shown destructive, degradative rearrangement and secondary apoptotic bodies (blue arrows) compared to the control group. APC treatment to OGD exposed cells leads to improvement in apoptotic features and improved morphologies of organelles, nuclei, and chromatin (Figure 2B). The cells were stained with annexin V-Allophycocyanin conjugated and PI to differentiate necrotic or terminal apoptosis cells from healthy growing cells. The flow cytometry examination revealed that OGD exposure caused early apoptosis in 25.37%, late apoptosis in 7.64%, and necrosis in 8.62% of SH-SY5Y cells. However, cells treated with APC in addition to OGD exposure result in reduced early apoptosis, late apoptosis, and necrosis in 16.63%, 3.23%, and 2.33%, respectively (Figure 2C,D).

### 3.3. Effect of APC on the Expression of EPCR, PAR1, and PAR3

The presence of EPCR, PAR1, and PAR3 are required for APC’s defensive action; we sought the expression of these receptors following OGD with or without APC. Flow cytometry analysis showed APC treatment to OGD exposed cells increases the cell surface expression of EPCR, PAR1, and PAR3 compared to OGD and the normal control (Figure 3A,B). OGD exposed cells had higher expression of all three receptor proteins but not at a significant level. The co-expression analysis showed that co-expression of EPCR, PAR1, and PAR3 dramatically increased in the APC treated group compared with the OGD and control groups (Figure 3C,D).

### 3.4. Effect of APC on Caspase-3 Activity and Mitochondrial Dysfunction

Cells exposed to OGD and concurrently treated with APC show reduced caspase-3 activity compared to the OGD group (Figure 4A). Fluorescent microscopy reveals that cells treated with OGD only shown an enhanced loss of mitochondrial transmembrane potential, while APC treatment shows an improved loss of mitochondrial transmembrane potential compares to OGD (0.85 ± 0.07 vs. 0.62 ± 0.03; OGD vs. APC; *p*
< 0.05). This suggests that the APC treatment inhibits mitochondrial dysfunction by inhibiting the loss of mitochondrial potential and acts as an anti-apoptotic agent (Figure 4B,C) validated by flow cytometry (Figure 4D,E). There was a significant shift of the cells, suggesting that APC attenuates loss in the mitochondrial membrane potential induced by OGD.

### 3.5. Expression of PARP-1 and AIF

Western blot experiments showed that the expression of BCL-2, Bax, and PARP-1 proteins in SH-SY5Y cells was affected by OGD exposure with and without APC. The cells exposed to OGD showed significantly increased expression of PARP-1 proteins (Figure 5A,B). Expression of Bax and BCL-2 was evaluated by western blot; thus, the ratio of Bax/BCL-2 was calculated to measure the anti-apoptotic effect of APC. SH-SY5Y cells exposed to only the OGD group show a significantly higher Bax/BCL-2 ratio compared to the control and cells treated with APC in addition to OGD (Figure 5A,C). To confirm the western blotting result, gene expression of *BCL-2*, *Bax*, *PARP-1*, and *AIF* was measured using quantitative qRT-PCR (Figure 5D). Results indicated that the expression of APC treatment attenuates the expression of *Bax*, *AIF*, and *PARP-1* relative to the OGD exposure and increases the expression of *BCL-2*. These data indicate that changes in the BCL-2, Bax, and PARP-1 proteins were due to changes in the transcriptional level.

## 4. Discussion

In this study, we presented experimental evidence for the underlying mechanism responsible for the neuroprotective effect of APC in ischemia injury. We found that APC inhibits PARP-1 cleavage and AIF expression, both of which are reported as contributing to ischemic injury, which leads to apoptosis.

The neurons in the penumbra remain viable for several hours and are the most likely to be salvageable, allowing researchers to pursue a number of means of achieving better functional recovery in stroke patients. To protect the ischemic neuronal cells due to irreversible ischemic injury, various therapeutic modalities were investigated to find new neuroprotective regimens. Many of these targeted the neuronal receptors to limit the release of excitatory neurotransmitters, free radical formation, and enhance the repair system [32]. APC exerts anticoagulant, anti-inflammatory, and anti-apoptotic activities through distinctly different mechanisms [10,33,34]. APC attenuates inflammation and provides cytoprotective signaling and reduces vascular damage and stress [5,14,34,35,36,37]. In our study, we found that APC treatment promotes anti-apoptotic activity and increases the expression of EPCR, PAR-1, and PAR-3, which is a crucial step in the APC-mediated protective pathway, as previously described [8,10,15,38].

The loss of mitochondrial potential, an important parameter of the functional integrity of mitochondria, decreases when exposed to OGD in SH-SY5Y cells, and it determines the fate of a cell during physiological stress or adverse conditions (whether cells survive or not) and can be used as an indicator of early apoptosis [39]. Our results indicate that APC attenuates the loss of mitochondrial potential and promotes cell survival. Cell viability was measured by the CCK-8 assay, which is dependent on an intact plasma membrane and cellular dehydrogenase activity. Because APC treatment blocks caspase-3 activation, protection from the loss of mitochondria membrane potential may be due to the inhibition of caspase-3 activation by APC. These results are in agreement with the previous study showing that APC treatment inhibited caspase-3 dependent apoptosis [10].

The present study demonstrated that APC mediates its anti-apoptotic role via PARP-1 and AIF. PARP-1 activation regulates the function of mitochondria through the cleavage of poly-ADP-ribose in the nucleus. These fragments are capable of inducing the release of AIF directly from the mitochondria [40,41]. Studies have shown that ischemic stroke leads to the production of reactive oxygen species that result in overstimulation of PARP-1, an important cause of cell death in in-vivo models of stroke [42,43]. In previous studies, it has been shown that PARP-1 inhibitors significantly improve cell survival [44,45,46]. In this study, the effects of APC on PARP-1 expression were observed after 24 h of reoxygenation following OGD in SH-SY5Y cells and attenuated the expression of PARP-1, suggesting that the anti-apoptotic effect of APC is also associated with the downregulation of PARP-1 expression in the in-vitro model of ischemic injury in the SH-SY5Y cell. Activation of PARP-1 is an important step towards caspase-independent cell death, which is mediated by AIF, which is released from mitochondria and translocated to the nucleus [40,41,47,48]. Findings indicate that the APC has a strong influence on the expression of PARP-1 and AIF for its anti-apoptotic activity. However, in this study, the protective effect of APC on different neuronal cell types could not be assessed, which is a limitation of this study.

## 5. Conclusions

To conclude, this study suggests that APC treatment could act as a neuroprotective agent against brain ischemic injury by attenuating apoptotic cascade and inflammation. The APC displayed anti-apoptotic activity by downregulating PARP-1, Bax, and caspase-3 activity and by restoring mitochondrial function. APC, which also has anti-coagulant properties together with a neuroprotective nature, could be a promising candidate to treat ischemic stroke conditions and may be a potential therapeutic strategy for brain ischemia.

## Figures and Tables

**Figure 1 brainsci-10-00959-f001:**
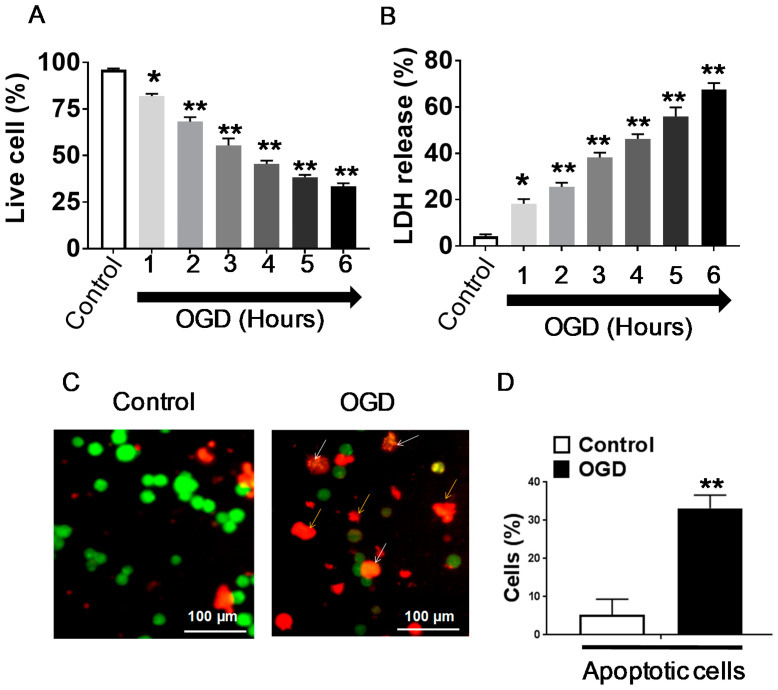
Cell Viability, toxicity, and morphological changes in differentiated SH-SY5Y cells following oxygen-glucose deprivation (OGD) exposure. (**A**) Assessment of SH-SY5Y cell survival following exposure to OGD was measured by Cell Counting Kit-8 (CCK-8) assays. (**B**) Assessment of cell injury by lactate dehydrogenase (LDH) assay. Cells were exposed to OGD for various time intervals. (**C**) Differentiated SH-SY5Y cells stained with a combination of acridine orange (100 μg/mL): ethidium bromide (100 μg/mL) 1:1 ratio. For OGD treatment, cells were exposed to 4 h of OGD. The white arrow shows necrosis and the yellow arrow show apoptotic cell death. (**D**) Representative graph showing the percentage of live cells and apoptotic cells control vs. OGD. Images were captured by an inverted fluorescent microscope (Nikon Instruments Inc., Tokyo, Japan) at 40× magnification. Six independent microscopic fields were selected for each experiment, and the data were expressed as mean ± SD for three independent experiments. Values were expressed as a percentage of the control. The data were expressed as the mean ± SD of three experiments. * *p* ≤ 0.05, ** *p* ≤ 0.001.

**Figure 2 brainsci-10-00959-f002:**
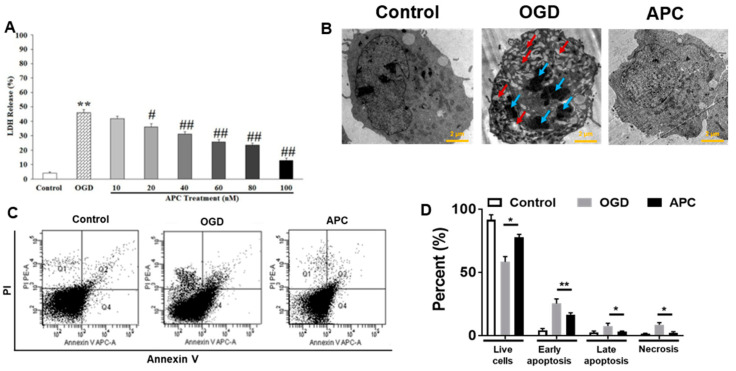
The Dose-dependent activity of activated protein C (APC) on toxicity and ultrastructure changes in SH-SY5Y cells after oxygen-glucose deprivation (OGD) exposure. The dose-dependent activity of APC: SH-SY5Y cells were treated with APC (10–100 nM) during and 24 h after the OGD exposure. (**A**) Effect of APC on lactate dehydrogenase (LDH) release% following OGD, which reveals that APC at 100 nM concentration provides maximum protection. (**B**) Cells were fixed and analyzed by transmission electron microscopy (TEM). Three samples in each group and six fields for each sample were examined. Bars = 2 µm. (**C**) Cells were stained with annexin V and propidium iodide (PI) 24 h after OGD exposure and then analyzed by flow cytometry. A representative dot plot of sorted cells by flow cytometry for each experimental condition is presented. (**D**) The pooled results of the percentages of live cells, early apoptotic cells, late apoptotic cells, and necrotic cells from three flow cytometry studies are presented. Viable cells = Q3, early apoptosis = Q4, late apoptosis = Q2, and necrosis = Q1. Analyses were done from three independent experiments by one-way ANOVA for the whole group and t-test between two groups. * *p* ≤ 0.05; ** *p* ≤ 0.001; # *p* ≤ 0.005 OGD vs. APC treatment; ## *p* ≤ 0.001 OGD vs. APC treatment.

**Figure 3 brainsci-10-00959-f003:**
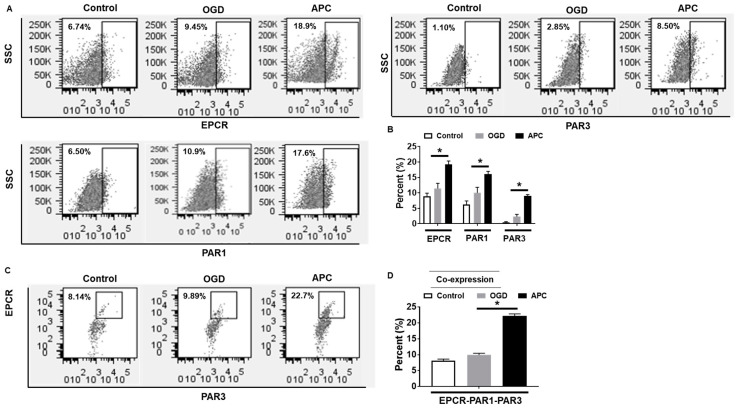
Flow cytometry analysis of endothelial cell protein C receptor (EPCR), protease-activated receptor (PAR)1, and PAR3 expression. (**A**) The expression of EPCR, PAR1, and PAR3 in the control, oxygen-glucose deprivation (OGD) exposed, and activated protein C (APC) in addition to the OGD group was quantified by flow cytometry. (**B**) Representative graph of flow cytometry for EPCR, PAR1, and PAR3. (**C**) The co-expression of EPCR, PAR1, and PAR3 in the control, OGD, and APC groups were quantified by flow cytometry. (**D**) Representative graph of Flow Cytometry for co-expression analysis of EPCR, PAR1, and PAR3. Data represented are the mean ± SD of three separate experiments. * *p* ≤ 0.05.

**Figure 4 brainsci-10-00959-f004:**
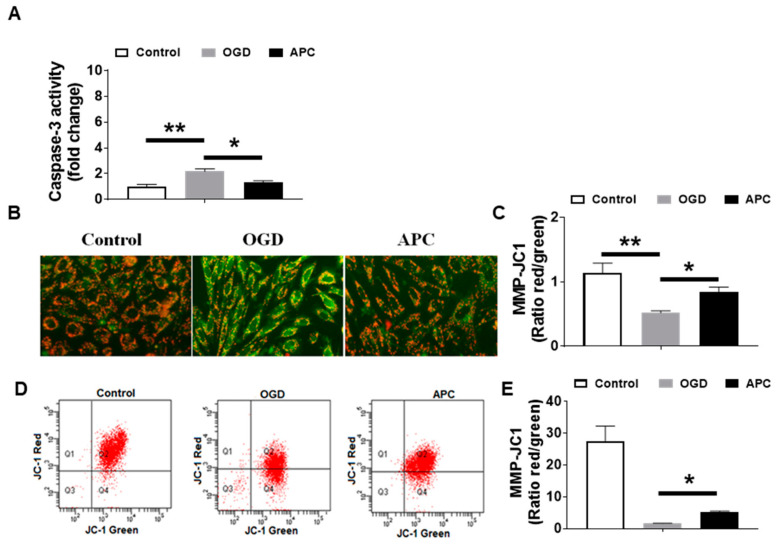
Caspase-3 activity and loss of mitochondrial membrane potential. (**A**) Caspase-3 activity was quantified by monitoring the hydrolysis of the corresponding caspase-3 specific colorimetric substrates. Induction of caspase-3 activity is observed in the oxygen-glucose deprivation (OGD) group. A significant increase in caspase-3 was observed while the activated Protein C (APC) treatment significantly attenuates the caspase-3 activity. (**B**) Representative fluorescence images of SH-SY5Y cells captured after 24 h (h) following OGD with and without APC (100 nM) stained with JC-1 and observed using fluorescent microscopy. (**C**) Graph showing the ratio of the intensity of red to green fluorescence (JC-1 aggregate and JC-1 monomer). (**D**) Loss of mitochondrial membrane potential was detected by flow cytometry. (**E**) Graph showing the ratio of the intensity of red to green fluorescence (JC-1 aggregate and JC-1 monomer) of flow cytometry. Data represented are the mean ± SD of 4 separate experiments. * *p* ≤ 0.05, ** *p* ≤ 0.001.

**Figure 5 brainsci-10-00959-f005:**
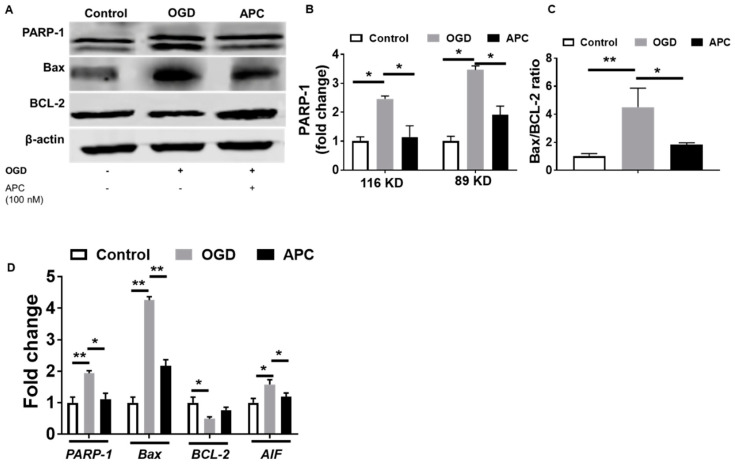
Changes in the expression of protein and mRNA. (**A**) Representative western blots showing the effect of activated protein C (APC) on B-cell lymphoma 2 (BCL-2), BCL-2-like protein 4 (Bax), and Poly(ADP-ribose) polymerase 1 (PARP-1) expression (*n* = 4). (**B**) The bar graph shows that APC down-regulated PARP-1. (**C**) Bar graph showing Bax/BCL-2 ratio following OGD exposure with and without APC. The expression of BCL-2, Bax, and PARP-1 was normalized to the expression of β-actin. (**D**) Relative mRNA expression of BCL-2, Bax, PARP-1, and apoptosis-inducing factor (AIF) in SH-SY5Y cells. The data represented are the fold changes ± SD of three separate experiments, * *p* ≤ 0.05; ** *p* < 0.01.

**Table 1 brainsci-10-00959-t001:** List of primer sequences for BCL-2-like protein 4(*Bax*), B-cell lymphoma 2 (*BCL2*), Poly(ADP-ribose) polymerase 1 (*PARP-1*), apoptosis-inducing factor (*AIF*), and glyceraldehyde 3-phosphate dehydrogenase (*GAPDH)* for quantitative real-time PCR (qRT-PCR).

Gene	Primer Sequence (5′-3′)
*BCL2*	F: GTGCCACCTGTGGTCCACCT
	R: CTTCACTTGTGGCCCAGATAGG
*Bax*	F: GCTTCAGGGTTTCATCCAGG
	R: AACATGTCAGCTGCCACTCG
*PARP-1*	F: TTGAAAAAGCCCTAAAGGCTCA
	R: CTACTCGGTCCAAGATCGCC
*AIF*	F: AAATCTCTCCACTACACT
	R: AATTTTAGCAGATTAAGAAGC
*GAPDH*	F: AACAGCGACACCCATCCTC
	R: CATACCAGGAAATGAGCTTGACAA

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
