# Peer review of "Neuroprotective Effects of Activated Protein C Involve the PARP/AIF Pathway against Oxygen-Glucose Deprivation in SH-SY5Y Cells"

_brainsci, 2020, doi:10.3390/brainsci10120959_

Round 1
Reviewer 1 Report
The authors are addressing an important area- identification and evaluation of potential neuroprotective agents for brain ischemia. The work carried out was to a relatively high standard and the results do show step change in neuoroprotective potential making this work of interest for Brain Sciences.
I would therefore recommend a thorough proof read and editing of english language and the ammendments outlined below.
Line 36-37: "The anticoagulant characteristics of APC have been well studied." The authors should cite high impact peer reviewed manuscripts demonstrating the well studied characteristics of APC in anticoagulation.
Figure 2A. Figure legends are inadequate and require better labelling and explanation. For the LDH release experiment no explanation for # sign is given.
Figure 2B. The manuscript would benefit from arrows directed to the TEM features described by the authors in the text to enable the reader to truly assess the images.
Line 280-281. "The cells exposed to OGD showed significantly decreased expression of BCL-2 protein." The result claimed here by the author does not reflect the western immunoblot presented in Figure 5A. In fact, it appears that BCL-2 expression is quite consistent throughout for both controls and treated samples. Furthermore no statistical analysis is performed on said western blot as such the authors cannot state that such protein expression is significantly decreased.
Author Response
Dear Reviewer,
Thank you for giving us to submit a revised draft of the manuscript “Neuroprotective effects of Activated Protein C involve the PARP/AIF pathway against oxygen-glucose deprivation in SH-SY5Y cells” for publication in the Brain Sciences Journal. We appreciate the time and effort that you invested in providing feedback on our manuscript and are grateful for the insightful comments on and valuable improvements to our paper. We have incorporated most of the suggestions made by the reviewers. These changes are done with Track Changes so could be identified easily in the manuscript. Please see below for a point-by-point response to the reviewers’ comments and concerns. All page numbers refer to the revised manuscript file with tracked changes.
I would therefore recommend a thorough proof read and editing of English language.
Answer: We went through the whole manuscript and rechecked by a native English speaker to improvise the manuscript.
Line 36-37: "The anticoagulant characteristics of APC have been well studied." The authors should cite high impact peer reviewed manuscripts demonstrating the well studied characteristics of APC in anticoagulation.
Response: We have added references (line 40) for anticoagulant characteristics of APC as follows:
- Jackson, C.; Whitmont, K.; Tritton, S.; March, L.; Sambrook, P.; Xue, M. New therapeutic applications for the anticoagulant, activated protein C. Expert Opin. Biol. Ther. 2008, 8, 1109–1122, doi:10.1517/14712598.8.8.1109.
- Dahlbäck, B.; Villoutreix, B.O. Regulation of blood coagulation by the protein C anticoagulant pathway: novel insights into structure-function relationships and molecular recognition. Arterioscler. Thromb. Vasc. Biol. 2005, 25, 1311–1320, doi:10.1161/01.ATV.0000168421.13467.82.
Figure 2A. Figure legends are inadequate and require better labelling and explanation. For the LDH release experiment no explanation for # sign is given.
Response: Thank you for pointing this out. We have explained and added the information for missed sign # (line 257-258)
Figure 2B. The manuscript would benefit from arrows directed to the TEM features described by the authors in the text to enable the reader to truly assess the images.
Response: We have added the suggested content to the manuscript in the text line 225-229. The TEM image was described and denoted accordingly in Figure 2B.
Line 280-281. "The cells exposed to OGD showed significantly decreased expression of BCL-2 protein." The result claimed here by the author does not reflect the western immunoblot presented in Figure 5A. In fact, it appears that BCL-2 expression is quite consistent throughout for both controls and treated samples. Furthermore no statistical analysis is performed on said western blot as such the authors cannot state that such protein expression is significantly decreased.
Response: To conclude the anti-apoptotic effect of activated protein C (APC) we analyzed the ratio of Bax/BCL-2 protein and found that it is highly upregulated in the oxygen-glucose deprivation group compared to control, while APC treatment significantly reduces the Bax/BCL-2 ratio. There was a typo error in the figure number, so it was different in the text and figure. The text is re-written (line 304-309) as per the figure and in figure 5C bar graph for Bax/BCL-2 ratio is incorporated instead of Bax and BCL-2 alone. The statistical analysis for mRNA expression is done in figure 5D.
Reviewer 2 Report
The Sriwastva’s et al, have done good work in the manuscript "Neuroprotective effects of Activated Protein C involve the PARP/AIF Pathway against oxygen-3 glucose deprivation in SH-SY5Y cells".
In the current study authors show that APC has potent protective effects against in vitro ischemia in SH-SY5Y cells by modulating mitochondrial function.
The manuscript has been well written, the results are clearly described,
Major comments
If another cell line can be used to further confirm this result, it will be more convincing and more conducive to the reader's understanding of the article.
Author Response
Dear Reviewer,
Thank you for giving us to submit a revised draft of the manuscript “Neuroprotective effects of Activated Protein C involve the PARP/AIF pathway against oxygen-glucose deprivation in SH-SY5Y cells” for publication in the Brain Sciences Journal. We appreciate the time and effort that you invested in providing feedback on our manuscript and are grateful for the insightful comments and valuable improvements to our paper.
Comments and Suggestions for Authors
Major comments
If another cell line can be used to further confirm this result, it will be more convincing and more conducive to the reader's understanding of the article.
Response: We agree with the reviewer’s assessment and suggestion. Unfortunately, this is the limitation of the study. We also tried with HCN-2 cell lines (human cortical neuronal cell), it was not successful in replicating the experiment because of HCN’s prolonged replication, growth, high doubling time, and low passage number.
Round 2
Reviewer 2 Report
There are many nerve cell lines, do not know why not it do? Hope to do this work in the future research.
Author Response
Dear Reviewer,
Thank you for giving me the opportunity to submit a revised draft of my manuscript titled “Neuroprotective effects of Activated Protein C involve the PARP/AIF pathway against oxygen-glucose deprivation in SH-SY5Y cells” to Brain Sciences Journal. We appreciate the time and effort that you have dedicated to providing your valuable feedback on my manuscript.
Comments and Suggestions for Authors
Comment: There are many nerve cell lines, do not know why not it do? Hope to do this work in the future research.
Response: Thank you for suggesting an important point. It would have been interesting to explore this aspect and we will definitely include other neuronal cell lines in future research studies. Additionally, we have made some minor corrections in methods (Blue and purple color) and re-write the conclusion section (line 372-376) as per your valuable suggestions.
This manuscript is a resubmission of an earlier submission. The following is a list of the peer review reports and author responses from that submission.